# Direct experimental observation of the molecular $J_{eff} = 3/2$ ground state in the lacunar spinel $GaTa_4Se_8$

Min Yong Jeong[1], Seo Hyoung Chang[2], Beom Hyun Kim[3,4], Jae-Hoon Sim[1], Ayman Said[5], Diego Casa[5], Thomas Gog[5], Etienne Janod[6], Laurent Cario[6], Seiji Yunoki[3,4,7,8], Myung Joon Han[1] & Jungho Kim[5]

Strong spin–orbit coupling lifts the degeneracy of $t_{2g}$ orbitals in $5d$ transition-metal systems, leaving a Kramers doublet and quartet with effective angular momentum of $J_{eff} = 1/2$ and $3/2$, respectively. These spin–orbit entangled states can host exotic quantum phases such as topological Mott state, unconventional superconductivity, and quantum spin liquid. The lacunar spinel $GaTa_4Se_8$ was theoretically predicted to form the molecular $J_{eff} = 3/2$ ground state. Experimental verification of its existence is an important first step to exploring the consequences of the $J_{eff} = 3/2$ state. Here, we report direct experimental evidence of the $J_{eff} = 3/2$ state in $GaTa_4Se_8$ by means of excitation spectra of resonant inelastic X-ray scattering at the Ta $L_3$ and $L_2$ edges. We find that the excitations involving the $J_{eff} = 1/2$ molecular orbital are absent only at the Ta $L_2$ edge, manifesting the realization of the molecular $J_{eff} = 3/2$ ground state in $GaTa_4Se_8$.

[1] Department of Physics, Korea Advanced Institute of Science and Technology, Daejeon 34141, Korea. [2] Department of Physics, Chung-Ang University, Seoul 06974, Korea. [3] Computational Condensed Matter Physics Laboratory, RIKEN, Wako, Saitama 351-0198, Japan. [4] Interdisciplinary Theoretical Science (iTHES) Research Group, RIKEN, Wako, Saitama 351-0198, Japan. [5] Advanced Photon Source, Argonne National Laboratory, Argonne, IL 60439, USA. [6] Institut des Matériaux Jean Rouxel (IMN), Université de Nantes, CNRS, 2 rue de la Houssinière, BP32229, 44322 Nantes cedex 3, France. [7] Computational Quantum Matter Research Team, RIKEN Center for Emergent Matter Science (CEMS), Wako, Saitama 351-0198, Japan. [8] Computational Materials Science Research Team, RIKEN Advanced Institute for Computational Science (AICS), Kobe, Hyogo 650-0047, Japan. Min Yong Jeong and Seo Hyoung Chang contributed equally to this work. Correspondence and requests for materials should be addressed to M.J.H. (email: mj.han@kaist.ac.kr) or to J.K. (email: jhkim@aps.anl.gov)

The quantum effects of electronic orbitals are pronounced in degenerate systems where the orbital degrees of freedom have to be considered on equal footing with spins as in, e.g., the Kugel–Khomskii model[1]. Examples include some cubic perovskite compounds of early 3d transition metals, in which the degeneracy of $t_{2g}$ orbitals is large and the oxygen octahedra are only weakly distorted. For heavy 5d electrons, the strong spin–orbit coupling (SOC) can reduce the degeneracy by splitting the $t_{2g}$ orbitals into a Kramers doublet ($J_{eff} = 1/2$) and quartet ($J_{eff} = 3/2$), and recovers the orbital angular momentum[2–4]. Recently, iridates with $5d^5$ have drawn much attention because the half-filled state near the Fermi level ($E_F$) is a Kramers doublet and a relatively weak electron correlation leads to the $J_{eff} = 1/2$ Mott ground state[5–7]. This state offers opportunities to explore quantum phases such as a topological Mott insulator[8], unconventional superconductivity[9–13] and quantum spin liquid[14–17].

At present, theories, modeling constructs and experimental investigations of relativistic $J_{eff}$ ground state systems are still emerging. Beyond the well-known $5d^5$ iridates, the main challenge in other 5d electron systems is to build a concrete understanding of the exotic quantum effects with the $J_{eff}$ state. A relatively simple, but more interesting case is the $5d^1$ system, which results in a $J_{eff} = 3/2$ effective moment. Examples can be found in double perovskites such as $Sr_2MgReO_6$[18], $Ba_2YMoO_6$[19–21] and $Ba_2NaOsO_6$[22, 23]. In the ionic limit, the magnetic moment of $J_{eff} = 3/2$ vanishes because the orbital component cancels the spin component[3, 4]. The spin–orbit entanglement leads to a strong multipolar exchange of the same order as the ordinary bilinear exchange[3, 4], giving access to a variety of exotic phenomena in multipolar systems such as 4f-/5f- heavy Fermion compounds[24]. While recent advanced X-ray spectroscopic studies showed clear signatures in $Sr_2IrO_4$ and the other iridates[6, 25–29] for the case of $J_{eff} = 1/2$, the physics of the $J_{eff} = 3/2$ state has to date remained elusive in experiment.

Recently, a lacunar spinel compound, $GaTa_4Se_8$, was suggested as a model system for the molecular $J_{eff} = 3/2$ Mott insulating ground state[30]. As shown in Fig. 1a, the basic building block is a tetramerized $Ta_4Se_4$, or simply so-called $Ta_4$ cluster. The short intra-cluster having a Ta–Ta distance of ≤3 Å naturally induces the molecular orbital (MO) states residing on the cluster[30, 31]. The MO calculation for the Ta–Ta bonding orbitals of $Ta_4$ cluster and the ab-initio band structure calculation found that one electron occupies the MO states with $t_2$ (or, $t_{2g}$-like) symmetry near $E_F$[30–35]. The strong SOC of the Ta atom splits the three-fold degenerate $t_2$ MO states into Kramers doublet ($J_{eff} = 1/2$ MO states) and quartet ($J_{eff} = 3/2$ MO states), and the quarter-filled state near $E_F$ is the Kramers quartet as shown in Fig. 1b. Due to the large inter-cluster distance (≥4 Å), the bandwidth of the band formed by $J_{eff} = 3/2$ MO states is small (~0.7 eV) and the relative strength of on-site Coulomb correlation, i.e., $U$ (~ 2 eV), is sizable, rendering $GaTa_4Se_8$ a rare example of a molecular $J_{eff} = 3/2$ Mott insulator[30].

Experimental identification of the relativistic $J_{eff}$ state is necessary to understand the underlying mechanisms of quantum phenomena that have been reported and speculated for this material and others[3, 18–23, 30, 32, 36–42]. At low temperature, for example, $GaTa_4Se_8$ exhibits an intriguing transition[38–40] towards a non-magnetic and possibly spin singlet state, which are presumably related to a peculiar bump observed in the susceptibility and specific heat[39, 40]. Furthermore, this non-trivial magnetic and electronic behavior could be related to superconductivity observed under pressure[32, 37, 38]. Considering that the previous studies do not take the SOC into account[30], determining the nature of its magnetic moment is essential to elucidate the physics of $GaTa_4Se_8$ and to address the related issues that have been theoretically discussed largely for the 5d oxides.

In the case of the $J_{eff} = 1/2$ ground state in $Sr_2IrO_4$, the salient experimental evidence has been that the magnetic resonant X-ray scattering (MRXS) intensity of the Néel-ordered state is nearly absent at the $L_2$ edge[6]. The destructive quantum interference at $L_2$ edge only occurs in the complex $J_{eff} = 1/2$ state ($\propto |xy, \mp\sigma\rangle \mp |yz, \mp\sigma\rangle + i|zx, \mp\sigma\rangle$), ruling out all single orbital $S = 1/2$ states of real wave functions. A few magnetically ordered iridium compounds were found to show the same phenomenon[26–29]. On the other hand, verifying the $J_{eff} = 3/2$ state in non-

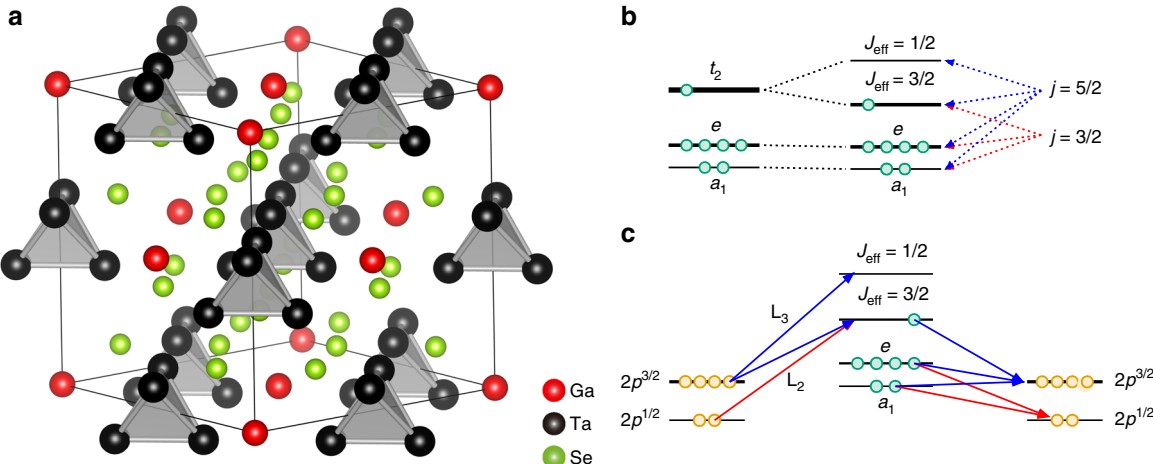

**Fig. 1** Crystal structure, MO levels, and RIXS process in $GaTa_4Se_8$. **a** The crystal structure of $GaTa_4Se_8$ (cubic $F\bar{4}3m$). The *red, black* and *green spheres* represent Ga, Ta and Se atoms, respectively. The $Ta_4$ tetrahedron clusters (shaded in *gray*) form a face-centered-cubic lattice. **b** The MO energy levels of $Ta_4$ cluster near the Fermi level ($E_F$). Due to the SOC, three-fold degenerate $t_2$ MO states split into Kramers quartet ($J_{eff} = 3/2$) and doublet ($J_{eff} = 1/2$) MO states. The former has the mixed character of the atomic $j = 3/2$ and $j = 5/2$, whereas the latter is branched off from the $j = 5/2$. **c** Schematic diagram for RIXS processes involving the $J_{eff} = 1/2$ and $J_{eff} = 3/2$ MO states. The low-energy dipole allowed non-elastic $L_2$- and $L_3$-edge RIXS processes are indicated by *red* and *blue arrows*, respectively. Ta 2p electrons in $p^{1/2}$ and $p^{3/2}$ core states are denoted by orange circles and 5d electrons occupying MO states near $E_F$ are represented by *green circles*. Since $2p^{1/2} \rightarrow J_{eff} = 1/2$ transition is forbidden, orbital excitations involving the $J_{eff} = 1/2$ MO states are absent in the $L_2$-edge RIXS. Therefore, only two elementary processes are allowed for the inelastic $L_2$-edge RIXS. This should be contrasted with the inelastic $L_3$-edge RIXS where five different processes are expected in low-energy excitations

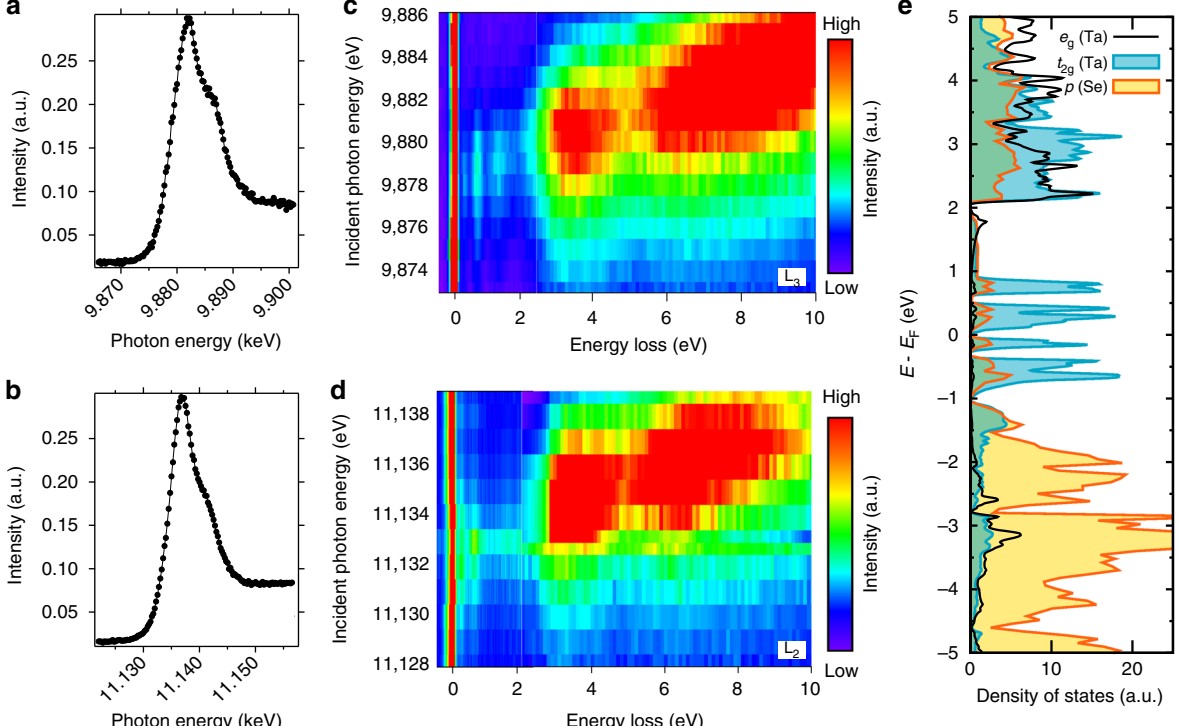

**Fig. 2** XAS and RIXS spectra and the projected DOS. **a**, **b** High-resolution Ta $L_3$-edge **a** and $L_2$-edge **b** XAS spectra of $GaTa_4Se_8$ measured in the partial yield mode. **c**, **d** High-resolution Ta $L_3$-edge **c** and $L_2$-edge **d** RIXS spectra as functions of the incident photon energy ($E_i$) and energy loss. Both spectra show the broad excitation over the three −7 eV energy loss. Below 2 eV, the three narrow peaks are clearly visible in the $L_3$-edge RIXS spectra. But the third peak at 1.3 eV is missing in the $L_2$-edge RIXS spectra. **e** The calculated DOS projected onto Ta-$e_g$ (black), Ta-$t_{2g}$ (cyan) and Se-$p$ (yellow) orbitals. In the region of $E_F$ ± 2 eV, the $t_{2g}$ states are dominant and responsible for the narrow peak excitations noticed in **c** and **d**

magnetic $GaTa_4Se_8$ is a greater challenge. MRXS analysis, which is only possible for magnetically ordered materials, cannot be exploited.

Here we used the high-resolution resonant inelastic X-ray scattering (RIXS) technique to explore the $J_{eff} = 3/2$ state. At the Ta L edges, dipole transitions give rise to the direct RIXS via $2p \rightarrow 5d$ absorption and subsequent $5d \rightarrow 2p$ decay, which directly probes the valence and conduction band states[43]. As depicted in Fig. 1b, c, the $J_{eff} = 1/2$ MO level is branched off from the atomic $j = 5/2$ and the absorption transition between $L_2$ ($2p^{1/2}$) and $j = 5/2$ is naturally dipole-forbidden[44]. In contrast, the $J_{eff} = 3/2$ MO states are composed of both the atomic $j = 5/2$ and $j = 3/2$ states. Therefore, we were able to establish the molecular $J_{eff} = 3/2$ ground state in $GaTa_4Se_8$ by examining the excitation spectra at both the Ta $L_3$ and $L_2$ absorption edges. This is the first spectroscopic evidence for a molecular $J_{eff} = 3/2$ ground state in a real material.

## Results

**Ta $L_3$ and $L_2$ edge XAS and RIXS.** Figure 2a, b show the high-resolution Ta $L_3$- and $L_2$-edge X-ray absorption spectroscopy (XAS) spectra, respectively, which were measured in the partial yield mode by recording the shallow core–hole emissions (see also Supplementary Fig. 1). The $L_3$- and $L_2$-edge spectra comprise one primary peak at ~ 9.8825 and ~ 11.1365 keV, respectively, and a shoulder peak at ~ 5 eV higher photon energy. Figure 2c, d show the high-resolution Ta $L_3$- and $L_2$-edge RIXS spectra, respectively, as a function of the incident photon energy ($E_i$). Both RIXS spectra show basically the same resonant profiles. It should be noted that broad excitation peaks ~3.5 and 7 eV are resonantly enhanced when $E_i$ is tuned near to the primary XAS peak. On the other hand, the narrow excitation peaks below 2 eV are

resonantly enhanced when $E_i$ is tuned to the ~ 2 eV below the XAS maximum.

The XAS and RIXS spectra clearly reveal the overall structure of the unoccupied states guided by insights from the band structure calculations. Figure 2e shows the wide energy range density of states (DOS), projected onto Ta atomic $t_{2g}$ and $e_g$ symmetry orbitals from the ab-initio band structure calculations. A quite large portion of the unoccupied states is located above 2 eV and has a mixed character of $t_{2g}$ and $e_g$ symmetry, which explains the overall XAS feature and the broad high-energy RIXS peaks above the 2 eV energy loss. Regarding these broad peaks, there is no distinct difference between the Ta $L_3$- and $L_2$-edge RIXS spectra.

On the other hand, the $t_{2g}$ symmetry character dominates the energy range near $E_F$ (±2 eV). In the XAS spectra, excitations to these $t_{2g}$ states, including the possible relativistic $J_{eff}$ states, do not show up as a distinct peak but are located in the lower-energy shoulder region of the large XAS peak. The narrow RIXS peaks below the 2 eV energy loss are assigned to orbital excitations within these $t_{2g}$ manifolds. In the case of the $L_3$-edge RIXS (Fig. 2c), three narrow peaks are located at 0.27, 0.7 and 1.3 eV energy loss positions. Remarkably, the 1.3 eV peak disappears in the $L_2$-edge RIXS spectra (Fig. 2d). In the sections below, to shed light on the physical origin, we further investigate the orbital excitation spectra in terms of the momentum transfer dependence, and analyze the RIXS spectra based on the band structure calculations and the cluster model calculations.

**The absence of 1.3 eV orbital excitation in the $L_2$ edge RIXS.** Figure 3a shows the momentum transfer dependence of the $L_3$-edge RIXS ($E_i = 9.879$ keV) excitations along (hhh) high-symmetry direction. Three orbital excitation peaks at the

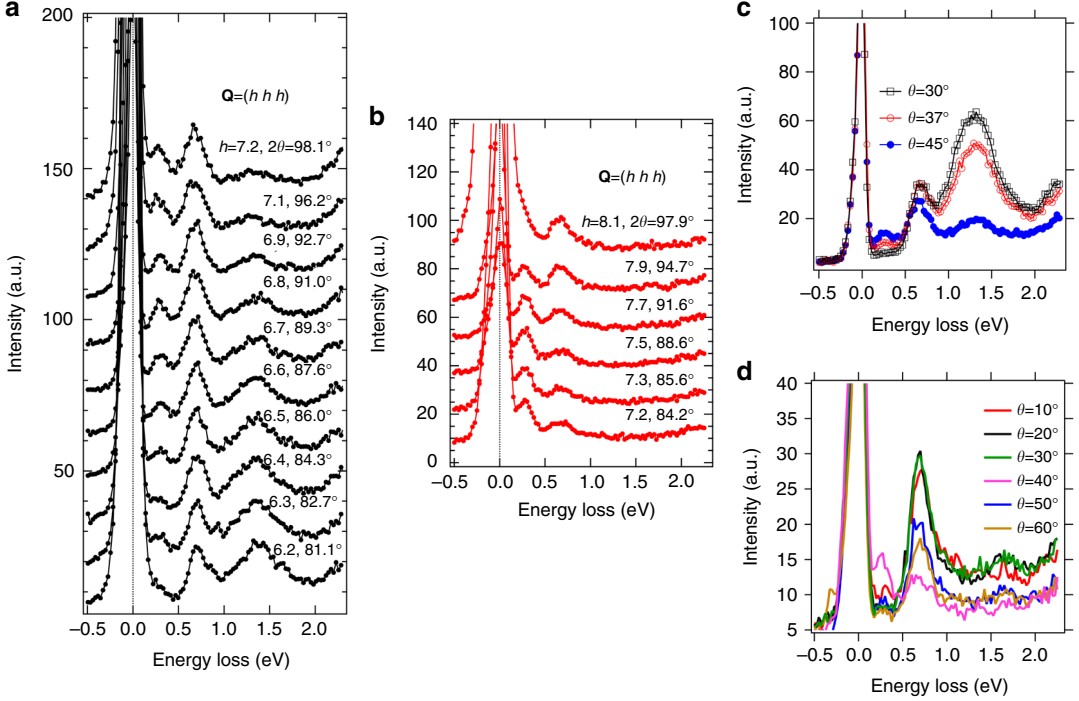

**Fig. 3** $L_3$- and $L_2$-edge RIXS spectra. **a, b** $L_3$-edge ($E_i = 9.879$ keV) **a** and $L_2$-edge ($E_i = 11.133$ keV) **b** RIXS spectra as a function of momentum transfer **Q** and energy loss. Three orbital excitations are clearly noticed at the 0.27, 0.7, 1.3 eV energy-loss positions in **a**. In sharp contrast, only two peaks at 0.27 and 0.7 eV are visible in **b**, and the broad peak at 1.3 eV is not observed. **c, d** $L_3$-edge **c** and $L_2$-edge **d** RIXS spectra as a function of incident sample angle $\theta$ and energy loss. $2\theta$ is fixed at 90°. Clearly, the broad peak at 1.3 eV is absent for $L_2$-edge

0.27, 0.7 and 1.3 eV energy loss positions are clearly identified for all momentum transfers with some intensity modulations. Within the instrument energy resolution (~ 100 meV), no dispersion is observed for these three excitations.

Figure 3b shows the momentum transfer dependence of the $L_2$-edge RIXS ($E_i = 11.133$ keV) excitations along the (hhh) high-symmetry direction. Like the $L_3$-edge RIXS excitations (Fig. 3a), two sharp peaks at the 0.27 and 0.7 eV energy loss positions are observed for all momentum transfers with some intensity modulations. Unlike the $L_3$-edge RIXS excitations, however, there is no peak at the 1.3 eV region in the $L_2$-edge RIXS excitations for all measured momentum transfers. This is our main observation which is attributed to the destructive interference of molecular $J_{eff} = 1/2$ state at the $L_2$ edge as will be further discussed below.

Figure 3c shows the incident sample angle ($\theta$) dependence of the $L_3$-edge RIXS excitations where the scattering angle ($2\theta$) is fixed to 90°. When a grazing angle ($\theta = 30°$) is used, the 1.3 eV peak is largely enhanced while the 0.27 eV peak is suppressed. Figure 3d shows the $\theta$ dependence of the $L_2$-edge RIXS excitations where $2\theta$ is fixed to 90°. For all $\theta$ angles, no peak structure shows up in the 1.3 eV energy loss region (also see Supplementary Fig. 2). Instead, a concave spectral shape is formed in the 1.3 eV energy loss region, indicating the total absence of the 1.3 eV peak.

**Band structure and cluster model RIXS calculations**. Having the solid experimental observation that there is no 1.3 eV peak only in the $L_2$-edge RIXS excitations, we now investigate the detailed electronic structure near $E_F$ and perform the cluster model RIXS calculations to find its origin. Figure 4a shows the schematics of the band structure near $E_F$ corresponding to the cases with and without SOC and electron correlation $U$. Without a sizable SOC, only a strong enough $U$ can split the $t_2$ MO band into a narrow lower Hubbard band (LHB) and a broad upper

Hubbard band (UHB). In this case, a broad orbital excitation between LHB and UHB is expected. Importantly, no contrast between the $L_3$- and $L_2$-edge RIXS spectra is expected. On the other hand, with a strong SOC, the well-defined $J_{eff} = 1/2$ MO band is branched off, leaving out the $J_{eff} = 3/2$ MO band near $E_F$. A moderate $U$ opens a gap, making it a molecular $J_{eff} = 3/2$ Mott insulator. Multiple orbital excitations are expected between the occupied bands ($a_1$, $e$ and $J_{eff} = 3/2$ LHBs), and the unoccupied $J_{eff} = 3/2$ and $J_{eff} = 1/2$ bands.

Figure 4b shows the calculated band dispersion (*right*) and DOS (*middle*) near $E_F$, which were projected onto the low-energy MO states ($a_1$, $e$, $J_{eff} = 3/2$, and $J_{eff} = 1/2$) formed in the $Ta_4$ tetrahedron cluster (see Fig. 1a, b). The band gap is formed within the $J_{eff} = 3/2$ MO bands, indicating the formation of a molecular $J_{eff} = 3/2$ Mott state. The unoccupied $J_{eff} = 1/2$ MO band is well separated from the $J_{eff} = 3/2$ MO bands. The DOS projected onto the Ta atomic $j = 5/2$ and 3/2 states is shown on the left of Fig. 4b. It demonstrates that the $J_{eff} = 1/2$ MO band comprises mostly the atomic $j = 5/2$ states, whereas the $a_1$, $e$ and $J_{eff} = 3/2$ MO bands are composed of both the atomic $j = 5/2$ and 3/2 states (see also Fig. 1b).

To clarify the nature of the observed excitations, we have carried out the cluster model calculations for RIXS spectra within the fast collision approximation (zeroth order of the ultrashort core–hole lifetime expansion) and the dipole approximation[45] (for details, see Supplementary Note 2). The calculated $L_3$-edge RIXS spectrum in Fig. 5a reveals four low-energy peaks. Peaks A and B originate from the excitations from the fully occupied $e$ and $a_1$ MO states to the partially occupied $J_{eff} = 3/2$ MO state, respectively. The excitations from the $e$ and $a_1$ MO states to the unoccupied $J_{eff} = 1/2$ MO state are well separated and comprise the remaining two peaks C and D, respectively (see Supplementary Note 4). The excitation from the $J_{eff} = 3/2$ MO state to the $J_{eff} = 1/2$ MO state is coincidently located at the second peak B. Figure 5b shows the corresponding experimental $L_3$-edge RIXS

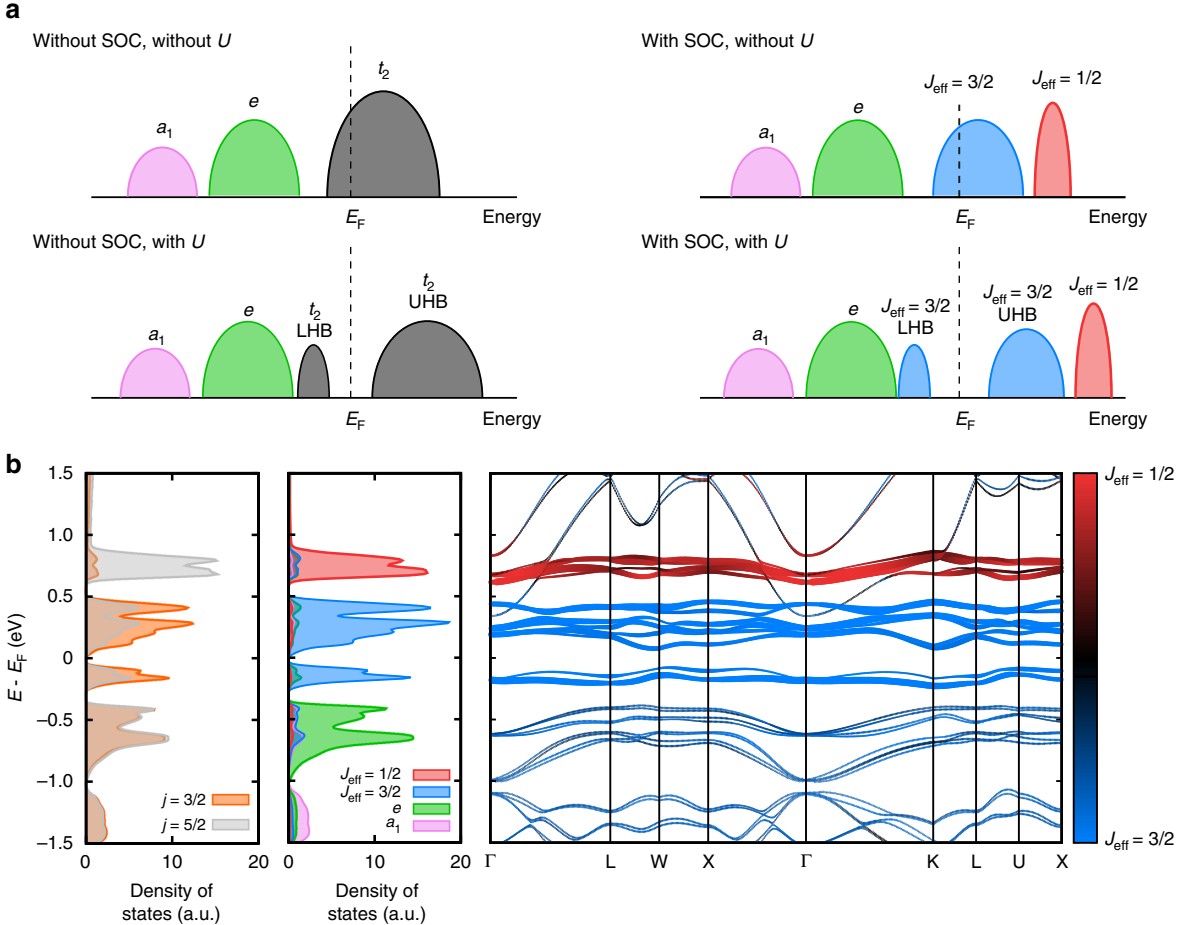

**Fig. 4** Schematic representation DOS and the electronic structure near $E_F$. **a** Schematic representation DOS for cases with and without SOC and electron correlation $U$. Without SOC and $U$, $t_2$ MO band prevails over $E_F$ with six-fold degeneracy. SOC (without $U$) splits this degenerate band into the four-fold degenerate $J_{eff} = 3/2$ and two-fold degenerate $J_{eff} = 1/2$ MO bands. On-site correlation $U$ (without SOC) can split the $t_2$ MO band into lower Hubbard band (LHB) and upper Hubbard band (UHB). With both SOC and $U$, the quarter-filled $J_{eff} = 3/2$ MO band splits to LHB and UHB with the higher-lying $J_{eff} = 1/2$ MO band. *Red, blue, green, pink* and *black* colors represent the $J_{eff} = 1/2$, $J_{eff} = 3/2$, $e$, $a_1$ and $t_2$ MO characters, respectively. **b** The calculated band dispersion (*right*) and DOS (*middle*) projected onto the MO states ($a_1$, $e$, $J_{eff} = 3/2$ and $J_{eff} = 1/2$) formed in the $Ta_4$ cluster, and the calculated DOS projected onto the Ta atomic $j = 3/2$ and $j = 5/2$ states (*left*). *Red, blue, green* and *pink* colors represent the $J_{eff} = 1/2$, $J_{eff} = 3/2$, $e$ and $a_1$ MO characters, respectively. In the band dispersion the $J_{eff} = 1/2$ and 3/2 character is also represented by the line thickness. *Gray* and *orange* colors indicate the $j = 3/2$ and $j = 5/2$ Ta atomic characters, respectively. Notice that the $J_{eff} = 1/2$ MO band is mostly composed of the atomic $j = 5/2$ states

spectra. We find reasonable agreement between the calculations and the experimental observations: A and B correspond to the first two peaks experimentally observed at 0.27 and 0.7 eV, and C and D correspond to the broad peak observed at 1.3 eV.

The excitations involving the unoccupied $J_{eff} = 3/2$ MO states (peaks A and B) are also revealed in the calculated $L_2$-edge RIXS spectrum in Fig. 5a. Compared to the $L_3$-edge RIXS calculations, the intensity of peak B is much weaker than that of peak A. This is understood because peak B contains the excitation from the $J_{eff} = 3/2$ MO states to the $J_{eff} = 1/2$ MO states, and its spectral weight is partially suppressed for the $L_2$-edge RIXS excitations in the following reason. For the RIXS process to occur, both photon absorption and emission must be the allowed transitions[43]. In the $L_2$-edge RIXS excitations, the photon absorption between $2p^{1/2}$ and the $J_{eff} = 1/2$ MO states is naturally dipole-forbidden because the $J_{eff} = 1/2$ MO states mostly comprise the Ta atomic $j = 5/2$ states as shown in Fig. 4b[44]. This is clearly seen in peaks C and D, which are totally absent in the calculated $L_2$-edge RIXS spectrum in the Fig. 5a, indicating that excitations involving the $J_{eff} = 1/2$ MO states was totally suppressed at the $L_2$ edge. Hence, the total absence of the 1.3 eV peak in the $L_2$-edge RIXS spectra of

$GaTa_4Se_8$ can be identified as arising from the destructive interference at the $L_2$ edge of the $J_{eff} = 1/2$ MO states, thereby establishing the molecular $J_{eff} = 3/2$ ground state in $GaTa_4Se_8$.

## Discussion

We have focused on spectroscopic evidence in search for the destructive quantum interference of $J_{eff}$ states. With the help of the band structure and the cluster model calculations, the RIXS excitation spectra taken at $L_3$ and $L_2$ edges provide clear evidence that the $J_{eff} = 1/2$ MO band is well separated from the $J_{eff} = 3/2$ MO band and the excitations involving the $J_{eff} = 1/2$ MO band are totally suppressed only at $L_2$ edge, verifying the molecular $J_{eff} = 3/2$ ground state in $GaTa_4Se_8$. Unlike MRXS, the RIXS technique can be useful even for a system with no long-range magnetic order (namely, a typical case rather than an exception) as demonstrated in the current study. Considering a strong SOC (~ 0.5 eV) of 5d orbital, this type of study is possible with a moderate energy resolution of ~ 100 meV, which is easily achievable for all 5d transition-metal L edges in the current state-of-the-art RIXS spectrometer[46].

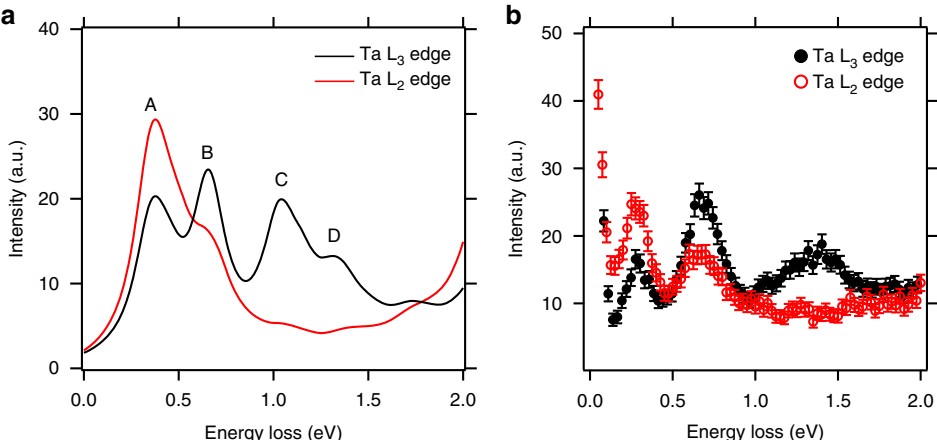

**Fig. 5** Cluster model calculations of the $L_3$ and $L_2$ RIXS spectra. **a** RIXS spectra calculated within the fast collision approximation (zeroth order of the ultrashort core–hole lifetime expansion) and the dipole approximation using the model parameters $U = 2$ eV, $\lambda_{SO} = 0.4$ eV, $t_\sigma = -1.41$ eV, $t_\delta = 0.213$ eV and $t_\pi = 0.1$ eV. $U$ and $\lambda_{SO}$ denote strengths of the on-site Coulomb repulsion and SOC, respectively, and $t_{\sigma,\delta}(t_\pi)$ denote diagonal (off-diagonal) nearest neighbor hoppings (see Supplementary Note 2). The spectral functions are convoluted with a Lorentzian function of 0.1 eV width. The lowest two peaks A and B are excitations from the fully occupied $e$ and $a_1$ MO states, respectively, to the partially occupied $J_{eff} = 3/2$ MO states. The excitation from the $J_{eff} = 3/2$ MO states to the unoccupied $J_{eff} = 1/2$ MO states is coincidently located at the second peak B. The excitations from the $e$ and $a_1$ MO states to the $J_{eff} = 1/2$ MO states are well separated and comprise the higher two peaks C and D, respectively. The latter three excitations involving the $J_{eff} = 1/2$ MO states are absent in the $L_2$-edge RIXS spectrum. **b** The corresponding experimental RIXS spectra at $L_3$- and $L_2$-edge with *error bars*. The *error bars* indicate the standard deviation to the number of detected photons

Establishing the molecular $J_{eff} = 3/2$ nature of GaTa$_4$Se$_8$ does not only just provide the opportunities for investigating $J_{eff}$ physics but also elucidates the current important issues in GaTa$_4$Se$_8$ and its close cousins such as GaNb$_4$X$_8$ and GaMo$_4$X$_8$ (X = Se, Te). In GaTa$_4$Se$_8$, for example, the underlying mechanism is not clearly understood for the 'paramagnetic' insulator to metal transition and the superconductivity under pressure[32–34], [37, 38]. Furthermore, the magnetic behavior of this material at low temperatures does not seem to support a simple 'paramagnetic' Mott phase[33, 34]. It should be emphasized that the relativistic $J_{eff}$ state has not been identified in the previous studies, since it is a recent theoretical finding[30] and is experimentally established in this study. On the basis of the current spectroscopic evidence, one can consider the ground state of GaTa$_4$Se$_8$ as the manifestation of the frustrated magnetic phase emerging from the non-trivial interactions among the relativistic $J_{eff} = 3/2$ moments[3, 19–21]. Moreover, we speculate that the superconductivity reported in this material is also related to this phase. We note that the other related lacunar spinel compound, GaNb$_4$Se$_8$ which is also expected to have the molecular $J_{eff} = 3/2$ nature[30], exhibits a quite similar low-temperature behavior and becomes superconducting. On the other hand, the molecular $J_{eff} = 1/2$ material with basically the same structure, GaMo$_4$X$_8$ (X = S, Se), is well understood as a ferromagnet[41] and does not exhibit superconductivity[47]. Our total energy calculation shows that the molecular $J_{eff} = 3/2$ moments of the Ta$_4$ cluster are in fact antiferromagnetically coupled between neighboring clusters ($E_{AFM-FM} = -7.4$ meV per cluster). Considering the fcc arrangement of this cluster unit, this strongly suggests magnetic frustration[48, 49]. In this regard, our current study may indicate that the molecular $J_{eff}$ moments are frustrated in this material. This is compatible with recent experimental observations of the specific heat and magnetic susceptibility, interpreted as a formation of dimerization and a spin singlet state[36, 39].

## Methods

**Partial-yield L-edge XAS.** Diced spherical analyzers were used to record L$_3$- and L$_2$-edge XAS spectra by analyzing resonant emission signals. The incident photon bandpass is ~0.8 eV. In the case of the L$_3$ edge, the L$_{\beta2}$ emission, which leaves out a

shallow (~ 230 eV) core–hole of 4$d$, was analyzed by the Ge (555) analyzer, which was on the 1 m Rowland circle. Because of a long lifetime of the shallow core–hole, a high-resolution (<2 eV) XAS was obtained[50]. In the case of the L$_2$ edge, the L$_{\gamma1}$ emission, which leaves out a shallow (~ 230 eV) core–hole of 4$d$, was analyzed by the Si (466) analyzer. Note that the use of the analyzer is essential for the L$_2$-edge XAS because a poor resolution of an energy-resolving detector cannot totally eliminate the Ga K-edge emission (~ 10.2 keV) from the Ta L$_{\gamma1}$ emission (10.9 keV).

**RIXS measurements.** The sample grown by the vapor transport method in a sealed quartz tube was mounted in a displex closed-cycle cryostat and measured at 15 K. The RIXS measurements were performed using the MERIX spectrometer at the 27-ID B beamline[46] of the Advanced Photon Source. X-rays were monochromatized to a bandwidth of 70 meV, and focused to have a beam size of 40 (H) × 15(V) μm$^2$. A horizontal scattering geometry was used with the incident photon polarization in the scattering plane. For the L$_3$-edge RIXS, a Si (066) diced spherical analyzer with 4 inches radius and a position-sensitive silicon microstrip detector were used in the Rowland geometry. For the L$_2$-edge RIXS, a Si (466) diced spherical analyzer with 4 inches radius was used. The overall energy resolution of the RIXS spectrometer at both edges was 100 meV, as determined from the full-width-half-maximum of the elastic peak.

**Sample synthesis.** Single crystal samples of GaTa$_4$Se$_8$ were obtained by the selenium transport method[51]. The pure powders of GaTa$_4$Se$_8$ were placed in an evacuated silica tube with a small excess of Se and heated at 950 °C for 24 h and then slowly cooled (2 °C h$^{-1}$) to room temperature.

**Band structure calculations.** Electronic structure calculations were performed by OpenMX software package[52], which is based on the linear combination of pseudo-atomic-orbital basis formalism. The exchange-correlation energy was calculated within the LDA (local density approximation) functional[53]. The energy cutoff The exchange-correlation energy was calculated within the LDA (local density approximation) functional of 400 Rydberg was used for the real-space integration and the 8 × 8 × 4 Monkhorst-Pack k-point grid was used for the momentum-space integration. The SOC was treated within the fully relativistic $j$-dependent pseudopotential and non-collinear scheme[54]. DFT + $U$ (density functional theory + $U$) formalism by Dudarev et al.[55, 56] was adopted for our calculations. Our main result is based on $U_{eff} = U-J = 2.3$ eV, and we found that our conclusion and discussion are valid for different $U_{eff}$ in a reasonable range (see Supplementary Fig. 4 and Supplementary Note 1). The experimental structure taken from X-ray diffraction[32] has been used for our calculation and there is no significant difference found in electronic and magnetic properties when the optimized structure is used. Total energy calculations have been performed with several different non-collinear magnetic configurations, and the most stable (a kind of antiferromagnetic) order has been taken to present the electronic structure. We found that the magnetic order does not change the band characters or their relative positions, and therefore, it does not affect any of our conclusion or discussion.

**Cluster model calculations**. We have adopted a three-band Hubbard model in a four-site tetrahedron cluster with seven electrons (for detail, see Supplementary Note 2). The model was solved numerically with the help of the Lanczos exact diagonalization method[57]. We have employed the Kramers–Heisenberg formula of the RIXS scattering operator[45] and calculated the RIXS spectra by using the continued fraction method[58]. The RIXS scattering operator was determined with the zeroth order of the ultrashort core–hole lifetime expansion and the dipole approximation was applied with taking the experimental X-ray beam geometry (see Supplementary Note 3).

**Data availability**. The data that support these findings are available from the corresponding authors (M.J.H. and J.K.) on reasonable request.

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

## Acknowledgements

We are grateful to Yang Ding, In Chung, Heung-Sik Kim, Jino Im, Hosub Jin and Michel van Veenendaal for helpful discussion. The use of the Advanced Photon Source at the Argonne National Laboratory was supported by the US DOE under Contract No. DE-AC02-06CH11357. M.Y.J., J.-H.S. and M.J.H. were supported by Basic Science Research Program through NRF (2014R1A1A2057202). The computing resource was supported by KISTI (KSC-2014-C2-046) and the RIKEN supercomputer system (HOKUSAI GreatWave). S.H.C. was supported by Basic Science Research Program through NRF (2016K1A3A7A09005337). B.H.K. was supported by the RIKEN iTHES Project. S.Y. was supported by Grant-in-Aid for Scientific Research from MEXT Japan under the Grant No. 25287096.

## Author contributions

S.H.C., M.J.H. and J.K. conceived and performed the experiment. A.S., D.C. and T.G. developed analyzers. E.J. and L.C. prepared the sample. M.Y.J., J.-H.S. and M.J.H. performed the band structure calculations. B.H.K. and S.Y. performed the cluster model calculations. All authors discussed the results. J.K. and M.J.H. led the manuscript preparation with contributions from all authors.

## Additional information

**Competing interests:** The authors declare no competing financial interests.

