## [Peer Review File · Nature Communications]

Reviewers' comments:

Reviewer #1 (Remarks to the Author):

This study reports the experimental verification of the novel $J_{\text{eff}} = 3/2$ state in GaTa_4Se_8 by means of RIXS.

By comparing the L3 and L2 RIXS spectra with DFT+U+SOC calculations the authors conclude that the absence of the

1.3 eV peak in the L2 spectrum, attributable to the fact that the $2p_{1/2}$ to $J_{\text{eff}}1/2$ transition is forbidden, is a clear fingerprint of

the J_{eff} nature of the ground state. In view of the intense interest on the novel type of magnetic and electronic states

realized in relativistic oxides this results is certainly interesting. Moreover, this appear to be the first application of RIXS to

detect this type of process in magnetically non-ordered materials.

However, it should be considered that this work represents an experimental verification of a previously reported result based on

first principles (Ref. 27, also published in Nat. Comm), which undermines the degree of physical novelty of this study.

From the point of view of a theoretical physics the experimental setup and the interpretation of the experimental data appear to be sound

and well-argued, but an expert opinion is certainly necessary. The first principles data, on the other hand, need some clarification,

as summarized below:

(1) In the main text the authors invoke a modest-U scenario but then base their quantitative analysis on the data obtained for $U_{\text{eff}} = 3$ eV,

which is not not a modest bur rather a large U. Previous studies have suggested a U of 0.6-1.2 eV (Ref. 31 and references therein).

According to Fig. S4, a smaller U would lead to a metallic solution, inconsistent with a relativistic Mott state. The J_{eff} picture would be somehow

still valid, from a qualitative point of view, but the conclusion on the excitation energies would be not valid anymore. This point should be carefully

address. If the authors want to use $U=3$ they need to justify this choice. If, as it seems, a small U appears more appropriate, they should re-elaborate

their interpretation. And, in any case, the discussion on the excitation energies is equivocal, see next point.

In passing, the author conclude that the system is a (relativistic) Mott insulator on the basis of DFT+SOC+U (line 206). Is it possible to reach this

conclusion from the RIXS data shown in Fig.2 and 3? While the L3 RIXS structure show a three peak structure with a rather low intensity right above the

Fermi energy, the L2 graph exhibits a more intense signal above the Fermi energy. If the RIXS data do indeed prove the existence of a Mott state, would it be possible

to provide an estimation of the band gap and compare it with previous data (0.12 eV). Also, a possible comparison between the RIXS and the optical conductivity

would also be interesting.

(2) Excitations. The interpretation of RIXS spectra based on DOS obtained by standard DFT calculations is a very simplified approach, and not really

well-suited, in particular considering the target impact of this work. Apart from the U problem, mentioned in the previous point, the most questionable

aspect is whether DFT (single-particle, no core-hole interaction, pseudopotential, no orbital relaxation, etc.) is capable to provide a quantitative

description of the RIXS process at all. A GW/BSE approach or a wavefunction-based scheme

employing a multi-determinant approach would be clearly the schemes to consider. In my opinion if the authors cannot provide a better computational description of the RIXS they should avoid to provide a quantitative interpretation based on DFT-DOS. The DFT DOS would only be functional in clarifying the Jeff state and the role of SOC and U in determining the sequence of Jeff states and in understanding qualitatively the consequence of the forbidden L2 excitation. To discuss the RIXS peaks the authors might use the energies given in Ref. 29, based on DFMT (0.6 and 1.2 eV), though in that study SOC effects seem to be neglected.

(3) Magnetism. The magnetic ground state of GaTa₄Se₈ is still debate but there is a general consensus about the absence of a long-range order. The computational set-up does not provide much information on the magnetic set-up used in the calculations. It seems that the authors have modeled the systems using a collinear long-range ordered FM and AFM state and found that the AFM ordering is more favorable. this is again a a questionable way of treating magnetic interaction in this system. First, the authors should explain why the supposedly paramagnetic state is model with an ordered magnetic setup; second, the strength of the spin-orbit coupling could give rise to non-collinear spin-structure; third, it is not clear how the magnetic ordering affects the electronic structure (apart from a vague statement on the robustness of the results with respect to AFM and FM spin ordering). General point: it would be more useful to include some details on the magnetic character of the system, earlier in the text.

(4) minor points. line 87: Ref. 27 does consider SOC; line 255: where does the given value of the SOC strength, 0.5 eV, come from?: line 281: this is a rather speculative statement, not really supported by the data shown in the paper.

In conclusion, the paper should undergo extensive revision before being considered for publication.

Reviewer #2 (Remarks to the Author):

I do not recommend the paper for publication at Nature Communications, as the scientific advance is quite incremental. It does not "represent important advances of significance to specialists" as required by the journal guideline. I would be glad to see the paper being published elsewhere, e.g. in Scientific Reports.

The main finding in the paper is experimentally proving the validity of the J_{eff} states, exactly as the title stated. The theoretical proposal of such a state was published previously as ref. 27. Notably, similar experimental methods have been used a few times in the investigation of iridates and osmates, and thus is not new. Thus, this paper produces neither vital theoretical knowledge nor experimental methods.

These being said, the paper itself is clearly written, and this paper is the first paper to try RIXS at Ta L-edges. However, it does not take new experimental methods to make RIXS possible at these edges.

Below I raise some of the questions, and encourage the authors to clarify them in future versions of the paper:

First, as stated in ref. 27, the J_{eff} state is built on a Ta₄ cluster. Meanwhile, RIXS is a local process based on a single Ta site. The dipole selection rule is only valid for the symmetry of the same entity. Note defining the angular momentum of a molecular cluster is different from that of a single atom. I understand here it is a coincidence (the two definitions are the same due to a relative phase of 0 between Ta atoms), but this should be made clear in the paper.

Second, it is quite dubious the J_{eff} states can account for the insulating behavior in the crystal in a way as the authors suggested. As clear from the crystal structure and stated in ref. 27, the Ta₄ clusters are far apart. The inter-(molecular) moment couplings can well be too weak to play a role. It is also quite a stretch to suggest a link between J_{eff} states and superconductivity.

Reviewer #3 (Remarks to the Author):

This manuscript reports on the study of the electronic structure of GaTa₄Se₄ by means of a resonant inelastic X-ray scattering experiment backed up by a band structure calculation. The authors convincingly demonstrate the $J_{\text{eff}}=3/2$ ground state in this 5d¹ system using a similar strategy as used in recent publications appeared on 5d⁵ iridates.

The result presented here is surely very interesting for the area of research on quantum effects of electronic orbitals, and sheds a new light on this class of materials. Moreover, the presentation of the experimental results is very clear and the study complete.

I am therefore strongly in favor of publication of this manuscript in Nature Communications.

Reply to the Referees Comments

We would like to thank Referees for their time and effort to review our manuscript. Thanks to the helpful comments by Referees, we believe that we have significantly improved the manuscript. The main improvement is the RIXS spectrum calculation, which is compared quantitatively with the experiments. This new and independent calculation strongly supports the main conclusion. In the following, we have tried to address all of the critiques and comments raised by Referees one-by-one.

Reply to the First Referee's Comments:

#0. (also related to the point #0 by Referee 2)

“In view of the intense interest on the novel type of magnetic and electronic states realized in relativistic oxides this results is certainly interesting. Moreover, this appear to be the first application of RIXS to detect this type of process in magnetically non-ordered materials.”

We thank Referee for correctly recognizing the novelty of our experiments. This is indeed the first successful application of a RIXS technique (to our best knowledge) to verify the quantum magnetic moment not being ordered in a conventional way.

“However, it should be considered that this work represents an experimental verification of a previously reported result based on first principles (Ref. 27, also published in Nat. Comm), which undermines the degree of physical novelty of this study.

We agree with Referee that our study would be of greater impact if it were not only the first experimental observation but also simultaneously the one containing the first theoretical prediction of a $J_{\text{eff}}=3/2$ ground state.

However, we would like to emphasize that the direct experimental observation of this kind of novel quantum moments has not been regarded as a small step. In the case of the most-studied Sr_2IrO_4 , a phase-sensitive MRXS (magnetic resonant x-ray scattering) measurement by B. J. Kim et al. (Science, 323, 1329, 2009) is widely accepted as a key step to confirm a $J_{\text{eff}}=1/2$ ground state even though its band character had already been

reported previously (B. J. Kim et al., PRL 101, 076402, 2008). Here we would also like to remind Referee of that many of new type ‘topological materials’ have first been predicted by band calculations and then confirmed by experiments, and these experimental observations (by, e.g., ARPES) have been usually considered as a major step and published in higher (or equal) profile journals.

Our current case of a molecular $J_{\text{eff}}=3/2$ ground state is a greater challenge in many regards even compared to the extensively studied $J_{\text{eff}}=1/2$ state. In particular, its ‘direct’ experimental detection is clearly more difficult because many candidate $J_{\text{eff}}=3/2$ materials do not exhibit any magnetic long-range order. Importantly, the simple expectations/predictions based on the naïve electronic structures have kept failing. In Table I, we list up the candidate $J_{\text{eff}}=3/2$ materials that have been studied before (to our best knowledge). While all of these materials are expected to have a $J_{\text{eff}}=3/2$ character, none of them has been confirmed experimentally in terms of effective moment, magnetic entropy, and magnetic order. For example, the DFT calculations supports a $J_{\text{eff}}=3/2$ ground state of $\text{Ba}_2\text{NaOsO}_6$, but it fails to account for the magnitude of the saturated moment, the observed anisotropy, or the negative Weiss temperature. Actually, the measured magnetic entropy change is rather consistent with a doublet ground state, not a quartet ground state[see S. Erickson *et al.*, PRL 99, 016404 (2007)]. We suspect that this may be one of reasons why previous studies have not proceeded to more elaborate experiments in searching for the evidence. Here we emphasize that the solid evidence should eventually come from the direct spectroscopic measurement as in the case of a $J_{\text{eff}}=1/2$ state. In this respect, our current study, unambiguously verifying a molecular $J_{\text{eff}}=3/2$ nature, is of crucial importance in this field of research; long been pursued but never been verified before.

To emphasize the challenge and importance of the direct experimental observation of a $J_{\text{eff}}=3/2$ ground state, we have changed the title of the manuscript and revised the main text in the revised version of the manuscript.

Table 1. A summary of previously-studied candidate materials for a $J_{\text{eff}}=3/2$ ground state.

Only indirect experimental measurements have been conducted to find that none of these materials satisfies the (simple) expectation for a $J_{\text{eff}}=3/2$ state. 'PM', 'FM', 'AFM', and 'SG' refers to paramagnetic, ferromagnetic, antiferromagnetic order, and spin glass phase respectively. The experimental data listed here can be found in the following reference: M. A. de Vries *et al.*, *PRL* 104, 177202 (2010); T. Aharen *et al.*, *PRB* 81, 224409 (2010); C. R. Wiebe *et al.*, *PRB* 65, 144413 (2002); K. Yamamura *et al.*, *J. Solid State Chem.* 179, 605 (2005); S. Erickson *et al.*, *PRL* 99, 016404 (2007); J. P. Carlo *et al.*, *PRB* 84, 100404(R) (2011); C. R. Wiebe *et al.*, *PRB* 68, 134410 (2003)

Compound	Effective Moment (μ_{eff})	Magnetic Entropy	Magnetic Order	DFT Calculations
Ba ₂ YMoO ₆	1.4 – 1.7	$\sim R \ln 4$	PM (>2K)	N/A
Ba ₂ LuMoO ₆	1.4	N/A	PM (>2K)	N/A
La ₂ LiMoO ₆	1.42	N/A	PM (>2K)	N/A
Ba ₂ LiOsO ₆	0.73	N/A	AFM (≤ 8 K)	N/A
Ba ₂ NaOsO ₆	0.6 – 0.68	$\sim R \ln 2$	FM (≤ 8 K)	$J_{\text{eff}}=3/2$
Ba ₂ CaReO ₆	0.74	$\sim R \ln 2$	AFM (≤ 15 K)	N/A
Sr ₂ MgReO ₆	1.72	$< R \ln 2$	SG (<50K)	N/A
Sr ₂ CaReO ₆	1.659	$< R \ln 2$	SG (<14K)	N/A

#1 & #2.

The following two points #1 and #2, we believe, are the main concerns of Referee. We thank Referee for raising these important issues. As Referee correctly pointed out, the interpretation of this kind of advanced spectroscopy experiments is often complicated and not straightforward, despite the fact that the first-principles band structure calculations very often provide a useful reference.

Well agreed with the Referee's concern, we have decided to perform the independent RIXS spectrum calculations since this is the first attempt of direct experimental observation of a $J_{\text{eff}}=3/2$ state and we also felt that the

more quantitative comparison between experiment and many-body theory is necessary. The results are included in the revised version of the manuscript. Two of us (B. H. K. and S. Y., who are newly included as co-authors) are the well-established experts on this kind of spectrum calculations, and have been working on RIXS and other spectroscopy calculations for iridates and other transition metal compounds.

The newly-included independent calculations of RIXS spectra clearly support our main conclusion that the suppressed +1.0~+1.3 eV peak in only L_2 edge is evidently attributed to the well-separated molecular orbital states ($a_1, e, J_{\text{eff}}=1/2$, and $J_{\text{eff}}=3/2$) and the formation of a molecular $J_{\text{eff}}=3/2$ ground state. The results are summarized in Figure 5 and the related discussion is mainly found in Page 12-14 in the revised version of the manuscript. Further details on the calculations can be found in Supplementary Notes 3, 4, and 5, and Supplementary Figure 5.

Although Referee does not ask this kind of direct calculations but simply raises questions regarding our (original) interpretation, the Referee's concerns and other ambiguities in the previous version of the manuscript have now been cleared up by these new calculations. Now our conclusion is more strongly supported, and we would like to thank Referee again for raising this point.

In the following, we reply to Referee's concerns about the interpretation based on the DFT calculations.

#1.

“(1) In the main text the authors invoke a modest-U scenario but then base their quantitative analysis on the data obtained for $U_{\text{eff}} = 3$ eV, which is not not a modest bur rather a large U. Previous studies have suggested a U of 0.6-1.2 eV (Ref. 31 and references therein). According to Fig. S4, a smaller U would lead to a metallic solution, inconsistent with a relativistic Mott state. The Jeff picture would be somehow still valid, from a qualitative point of view, but the conclusion on the excitation energies would be not valid anymore. This point should be carefully address. If the authors want to use $U=3$ they need to justify this choice. If, as it seems, a small U appears more appropriate, they should re-elaborate their interpretation. And, in any case, the discussion on the excitation

energies is equivocal, see next point.”

First of all, we would like to point out that the U values of 0.6 – 1.2 eV in Ref. 31 (Ref. 32 in the revised manuscript) that Referee refers to has different physical meaning from the U value in our manuscript. As Referee might be well aware of, our U is assigned to the entire Ta-5d orbitals in the standard DFT+U procedure. On the other hand, 0.6 – 1.2 eV has been used for the model studies (including DMFT calculation by Camjayi, Rozenberg *et al.*) in which only molecular ‘*t*’ manifolds are concerned as the ‘correlated subspace’. Additional screening of electronic correlations in molecular manifolds should be allowed due to large spatial extension of orbitals. Therefore it is not surprising but rather natural to use larger U to obtain the reasonable band structure. In fact, if we look at the GGA band dispersion, the bandwidth (W) of molecular ‘*t*’ is $W^t \sim 0.8$ eV and that of Ta- t_{2g} bandwidth is $W^{t_{2g}} \sim 2.5$ eV. Thus, it is not an artifact or based on an unreasonable physical assumption, but merely reflects the different theoretical approaches.

One can also put it in a slightly different way: The atomic Ta-5d orbitals are more localized than the molecular ‘*t*’ orbitals which are composed of the linear combinations of Ta- t_{2g} at four different sites. In fact, our cRPA (constrained random phase approximation) calculations find that the U value for molecular ‘*t*’ is ~ 0.7 eV and the U values for typical Ta or Ir compounds are $\sim 2.0 - 3.0$ eV. Considering Referee’s point, we updated the related discussion. See the revised Supplementary Notes.

In addition, we would like to note that our U value can be reduced down to ~ 2.3 eV without affecting any of our conclusion and discussion. Considering the Referee’s concern, in the revised version of the manuscript, all of the main calculations have been updated for $U=2.3$ eV, and the related discussion has also been added (see Figure 2 and 4).

“In passing, the author conclude that the system is a (relativistic) Mott insulator on the basis of DFT+SOC+U (line 206). Is it possible to reach this conclusion from the RIXS data shown in Fig.2 and 3? While the L3 RIXS structure show a three peak structure with a rather low intensity right above the Fermi energy, the L2 graph exhibits a more intense signal above the Fermi

energy. If the RIXS data do indeed prove the existence of a Mott state, would it be possible to provide an estimation of the band gap and compare it with previous data (0.12 eV). Also, a possible comparison between the RIXS and the optical conductivity would also be interesting.”

A short answer is that our RIXS measurement alone cannot lead to the conclusion that the system is a Mott insulator.

The charge excitation can be best studied by the indirect RIXS such as K-edge RIXS, where the charge excitation is induced via the shake-up process by the strong core-hole potential. In fact, the charge excitations probed by the RIXS can be compared with the optical data. [see J. Kim, et al.. PRB **79**, 094525 (2009)]

However, as evidenced in a number of the L-edge RIXS studies [see Ament, L. J. P., et al. *Rev. Mod. Phys.* **83**, 705 (2011)], the charge excitations are rather weak and the L-edge RIXS spectra are dominated by the local orbital and spin excitations such as d-d excitations and single-magnon excitations. In the case of the L-edge RIXS, the excited electron effectively screens the core hole and there is relatively little shake-up excitation. As a relevant example, please see the L₃ edge RIXS data of Sr₂IrO₄ [J. Kim, et al., PRL. **108**, 177003 (2012).], where the particle-hole excitations across the Mott gap is buried under the strong local orbital excitations.

As such, in our L-edge RIXS measurement, the charge excitations are buried by d-d excitations. In principle, the charge excitations across the Mott gap in the current system can be proved by the L₁ or K edge RIXS measurements, which is obviously out of the scope of the current study.

#2.

“(2) Excitations. The interpretation of RIXS spectra based on DOS obtained by standard DFT calculations is a very simplified approach, and not really well-suited, in particular considering the target impact of this work. Apart from the U problem, mentioned in the previous point, the most questionable aspect is whether DFT (single-particle, no core-hole interaction, pseudopotential, no orbital relaxation, etc.) is capable to provide a quantitative description of the

RIXS process at all. A GW/BSE approach or a wavefunction-based scheme employing a multi-determinant approach would be clearly the schemes to consider. In my opinion if the authors cannot provide a better computational description of the RIXS they should avoid to provide a quantitative interpretation based on DFT-DOS. The DFT DOS would only be functional in clarifying the Jeff state and the role of SOC and U in determining the sequence of Jeff states and in understanding qualitatively the consequence of the forbidden L2 excitation.

To discuss the RIXS peaks the authors might use the energies given in Ref. 29, based on DFMT (0.6 and 1.2 eV), though in that study SOC effects seem to be neglected.”

This important point concerns the reliability of our interpretation of the RIXS experiments based on the DFT band structure, especially in the quantitative aspects. Since we basically agree with Referee, we have decided to perform the RIXS spectrum calculations using the standard configuration-interaction method that can treat many-body multiplet effects. While each theoretical approach has its own limitations and advantages, we do not know any better theoretical reference than being obtained from this kind of direct spectrum calculations (GW/BSE is not well suited for this strongly correlated Mott insulator). With the new calculations, we were able to identify the nature of the low-energy excitations observed in the RIXS experiments. The details of the RIXS spectrum calculation can be found in the ‘Method’ part and Supplementary Notes in the revised manuscript. Now having both DFT-based electronic structure and the calculated RIXS spectrum, our interpretation is much more strongly supported, thereby confirming our main conclusion.

We thank Referee for this helpful comment. The main revision regarding this point can be found in Fig. 5 and in Page 12-14.

#3.

“(3) Magnetism. The magnetic ground state of GaTa4Se8 is still debate but there is a general consensus about the absence of a long-range order. The computational set-up does not provide much information on the magnetic set-up used in the calculations. It seems that the authors have modeled the systems

using a collinear long-range ordered FM and AFM state and found that the AFM ordering is more favorable. this is again a a questionable way of treating magnetic interaction in this system. First, the authors should explain why the supposedly paramagnetic state is model with an ordered magnetic setup; second, the strength of the spin-orbit coupling could give rise to non-collinear spin-structure; third, it is not clear how the magnetic ordering affects the electronic structure (apart from a vague statement on the robustness of the results with respect to AFM and FM spin ordering). General point: it would be more useful to include some details on the magnetic character of the system, earlier in the text.”

Firstly, we would like to make it clear that our DFT results are based on the non-collinear DFT+SOC+U calculation with the most stable spin order assumed after comparing the total energies of several possible non-collinear configurations. This comparison is already partly included and discussed in our previous publication Ref. 27 (Ref. 28 in the revised manuscript). Thus, the Referee’s second concern is already reflected in our calculation. Considering the Referee’s concern, we revised the manuscript to be more explicit about this point (see ‘Methods’ section and also ‘Discussion’ section).

The first and third points raised by Referee are closely related to each other. The reason that we assume a certain type of long-range magnetic order in our DFT calculations is first because there is no better way to deal with a paramagnetic insulating phase within the static Hartree-Fock DFT+U scheme. While the ground state magnetic configuration of this material is still unknown and under debate as Referee mentioned, we note that this material has a well-developed ‘Mott’ gap. In this case, the magnetic order does not affect the qualitative band character and the relative position of the bands . Thus the peak assignment and the overall RIXS interpretation are not affected by the assumed magnetic order.

We would like to note that our experiment has been conducted at temperatures well above the magnetic scales. Furthermore, our newly-updated spectrum calculations do not assume any type of long range magnetic order. Therefore, in any case, our main conclusion is not seriously harmed by the long-range magnetic order assumed in DFT calculations. We

would like to emphasize that the two different theoretical approaches strongly support our first experimental observation of a molecular $J_{\text{eff}}=3/2$ state.

Considering the Referee's comment, we have revised the relevant part of our manuscript to be clearer. See Figure 5, the section 'Methods' and 'Band structure calculations, cluster model RIXS calculations, and the absence of excitations involving $J_{\text{eff}} = 1/2$ MO states'.

#4.

“(4) minor points. line 87: Ref. 27 does consider SOC; line 255: where does the given value of the SOC strength, 0.5 eV, come from?: line 281: this is a rather speculative statement, not really supported by the data shown in the paper.”

We would like to thank Referee for pointing out possibly misleading expressions and sentences in the manuscript. We have revised them all in the new version of the manuscript. The SOC strength of ~ 0.5 eV is a widely accepted estimation. See, for example, Kim *et al.* *PRL* **108**, 177003 (2012); Witczak-Krempa *et al.*, *Annu. Rev. Cond. Matter Phys.* **5**, 57 (2014); Martins *et al.*, *PRL* **107**, 266404 (2011); Watanabe *et al.*, *PRL* **105**, 216410 (2010); Ou and Wu, *Sci. Rep.* **4**, 4609 (2014); Chikara *et al.*, *PRB* **92**, 081114(R) (2015); Ye *et al.*, *PRB* **87**, 140406 (R) (2013); Uematsu *et al.*, *PRB* **92**, 094405 (2015).

Reply to the Second Referee's Comments:

#0. The similar concern is also briefly mentioned by Referee 1. Please see our reply to Referee 1 (point #0).

“The main finding in the paper is experimentally proving the validity of the J_{eff} states, exactly as the title stated. The theoretical proposal of such a state was published previously as ref. 27. Notably, similar experimental methods have been used a few times in the investigation of iridates and osmates, and thus is not new. Thus, this paper produces neither vital theoretical knowledge nor experimental methods.”

With all our respect, we would like to add few more comments.

The main claim of our study is not merely about a J_{eff} state in general, but about first experimental observation of a $J_{\text{eff}}=3/2$ state. We would like to emphasize that $J_{\text{eff}}=3/2$ is a qualitatively different physical object from $J_{\text{eff}}=1/2$. A $J_{\text{eff}}=3/2$ state has been actively pursued but never been found or established until this study [please see the Table above and our arguments there (#0 for Referee 1)]. This is partially because there is no well-established $J_{\text{eff}}=3/2$ material ever reported. Even at the ‘lower level’ of indirect experimental investigations, none of the candidate materials satisfies the expected criteria. As shown clearly in the literature of $J_{\text{eff}}=1/2$ iridates, the direct evidence is of crucial importance and should come from the phase-sensitive spectroscopic technique. In fact, this is the very reason why this kind of direct experimental observation reported in this manuscript is always quite challenging.

There are a number of RIXS studies on iridates and few RIXS studies on osmates. But any study among those did not use the RIXS as a phase-sensitive spectroscopic technique to rigorously establish the J_{eff} state to the best of our knowledge. All RIXS spectra were taken only at the Ir/Os L_3 edge, mostly focusing on on the magnetic excitation spectra. Hence, we do not agree with the criticism that similar experimental methods have been used a few times in the investigation of iridates and osmates.

In order to highlight our main claim, we have changed the title of the manuscript to “Direct experimental observation of the molecular $J_{\text{eff}}=3/2$ ground state in a lacunar spinel GaTa_4Se_8 ”. Moreover, considering the Referee’s comment, we have revised the relevant sentences in the main text so that our main claim about a $J_{\text{eff}}=3/2$ state could be clearer. Since this is the first attempt for direct experimental observation of a $J_{\text{eff}}=3/2$ state, we have also decided to calculate the RIXS spectra using the configuration-interaction method, treating properly many-body multiplet effects, to make quantitative comparison with the experimental observation, which further strengthens our main conclusion.

#1.

“First, as stated in ref. 27, the J_{eff} state is built on a Ta_4 cluster. Meanwhile, RIXS is a local process based on a single Ta site. The dipole selection rule is only valid for the symmetry of the same entity. Note defining the angular momentum of a molecular cluster is different from that of a single atom. I understand here it is a coincidence (the two definitions are the same due to a relative phase of 0 between Ta atoms), but this should be made clear in the paper.”

We would like to thank Referee for pointing this out. We agree that the description was not clear enough in the previous version of the manuscript. As Referee correctly points out, RIXS basically measures the atomic spectra and therefore it is sensitive to the local atomic symmetry. However, one can apply the same symmetry argument for molecular orbitals in this system because the relevant low-energy molecular orbital states are composed only of Ta- t_{2g} orbitals. To avoid this unnecessary confusion, in the revised version of the manuscript, we have used different terminologies to describe the molecular orbital and the atomic orbital, and revised the way to explain why the selection rule can still apply in this molecular based system. We have also revised the schematic figures in Figs. 1(b) and 1(c). We believe that the revised version of the manuscript is much clearer in this regard.

Furthermore, in the revised version of the manuscript, we have performed the independent RIXS spectrum calculations using the configuration-interaction method with correctly considering both atomic and molecular orbital symmetries (see above; Reply to Referee 1), which also strongly supports our claim.

#2.

“Second, it is quite dubious the J_{eff} states can account for the insulating behavior in the crystal in a way as the authors suggested. As clear from the crystal structure and stated in ref. 27, the Ta_4 clusters are far apart. The inter-(molecular) moment couplings can well be too weak to play a role. It is also quite a stretch to suggest a link between J_{eff} states and superconductivity.”

We agree with Referee that the J_{eff} state cannot account for the insulating behavior of this material in the sense that the inter-cluster moment coupling

induces the insulating ground state. As the previous studies suggest, the intra-cluster Hubbard U , which is much larger than the inter-cluster hopping, is responsible for the Mott insulating gap. This is very similar to a family of organic molecular conductors [see, e.g., Kanoda, J. Phys. Soc. Jpn. **75**, 051007 (2006)]. While there may be still room to re-investigate this currently prevailing understanding, it is certainly not our main interest of current study and is not directly related to any of our main conclusion.

Regarding the superconductivity, we believe that Referee concerns the second paragraph in Discussion Section. Note that the purpose of this paragraph is to discuss the possible implications of our new finding and to present the perspective for possible future directions of this material research. Thus it is speculative by nature. It is clear that our study in this manuscript is not about the relation between $J_{\text{eff}}=3/2$ moment and superconductivity. However, we believe that it is important and highly interesting to examine possible unconventional superconductivity that may be induced once mobile carrier is introduced into the $J_{\text{eff}}=3/2$ state, since this should be contrasted with, for example, high- T_c cuprate superconductors (superconductivity induced in a spin $S=1/2$ Mott insulator), possible superconductivity in Sr_2IrO_4 (superconductivity induced in a $J_{\text{eff}}=1/2$ Mott insulator), and molecular conductors where superconductivity is induced in frustrated Mott insulators. After all, GaTa_4Se_8 is the first material in which the emergence of a $J_{\text{eff}}=3/2$ ground state is experimentally confirmed by our current study and has been reported to be superconducting under pressure.

Considering the Referee's comment, we have revised our discussion not to be misleading (see Page 15, Paragraph 2 (the last paragraph of 'discussion' section)).

In the revised version of the manuscript, we have substantially revised the manuscript to strengthen our main conclusion and hopefully Referee now will find our study suitable to the publication in Nature Communications.

Reply to the Third Referee's Comments:

We would like to thank Referee for the careful review and assessment. In particular, we are very glad to know that Referee correctly identifies our study being about the ' $J_{\text{eff}}=3/2$ ground state' (distinctive from the case of $J_{\text{eff}}=1/2$), and 'strongly' recommends the publication. We are also very pleased that Referee finds our presentation 'very clear and the study complete'.

REVIEWERS' COMMENTS:

Reviewer #1 (Remarks to the Author):

The authors have addressed satisfactorily all issues raised in my previous reports. In particular, they have conducted a new analysis on the excitations based on model cluster calculations, which should provide more sound insights on the excitation process compared to DFT. Second, they have clarified the choice of the Coulomb parameter U , and decided to present results obtained for $U_{\text{eff}}=2.3$ instead of 3.0 eV (and actually also in the RIXS simulations the authors have used $U=2$ eV, a further indication that the value $U_{\text{eff}}=3$ used in the original version is much too large).

Regarding the choice of U , the authors mention in the letter "our cRPA calculations find that the U value for the molecular 't' orbitals ~ 0.7 eV"; it is natural to ask which value they would obtain by including the full d band.

Reviewer #2 (Remarks to the Author):

I would like to thank the authors for addressing some of my concerns. My concern #2 (on how J_{eff} state relates to other properties of the material) is now clarified. My concern #1 consists of two parts: (1) the nature of the J_{eff} state (from a molecular cluster) – this is now properly discussed; and (2) how RIXS (a probe localized to a single atom) could explore the state of a molecule. This is not explicitly clarified earlier in the paper, but I would do with the new RIXS calculation (Fig. 5).

These being said, I still would not recommend the paper for publication, as I firmly believe the advances in this paper is incremental and do not live up to the standards of Nature Communications. The principle of using resonant X-rays to clarify the nature of J states is used in a series of papers in a range of journals (B. J. Kim et al., Science, 323, 1329 (2009); J. Kim et al., Phys. Rev. Lett. 108, 177003 (2012); S. Calder et al., Nature Communications 7: 11651 (2016); S. Calder et al., Phys. Rev. B 95, 020413(R) (2017)). As the authors agreed in the rebuttal letter, the J states is not a new notion, and the technical path was well laid out in previous papers. I could not agree with the "challenge and importance of the direct experimental observation" that the authors stated in the rebuttal letter as a reason for publication.

Meanwhile, I would recommend the paper for Sci. Reports or other field-specific journals.

Reply to the Referees Comments

We would like to thank Referees for their time and effort to review our manuscript.

Reviewer #1 (Remarks to the Author):

The authors have addressed satisfactorily all issues raised in my previous reports. In particular, they have conducted a new analysis on the excitations based on model cluster calculations, which should provide more sound insights on the excitation process compared to DFT. Second, they have clarified the choice of the Coulomb parameter U , and decided to present results obtained for $U_{\text{eff}}=2.3$ instead of 3.0 eV (and actually also in the RIXS simulations the authors have used $U=2$ eV, a further indication that the value $U_{\text{eff}}=3$ used in the original version is much too large).

Regarding the choice of U , the authors mention in the letter "our cRPA calculations find that the U value for the molecular 't' orbitals ~ 0.7 eV"; it is natural to ask which value they would obtain by including the full d band.

Reply:

We would like to thank Referee for the careful review and assessment. In particular, we are very glad to know that Referee appreciates the new analysis based on model cluster calculations and the clarification of the choice of the Coulomb parameter U in our revised manuscript. As for the U value for 'full d orbitals', or atomic d orbitals, we tried cRPA calculation but found that it is not easy to numerically identify a specific orbital, indicative of the molecular orbital nature. Although we cannot present the calculated value, it is natural to expect a larger value as we discussed before.

Reviewer #2 (Remarks to the Author):

I would like to thank the authors for addressing some of my concerns. My concern #2 (on how J_{eff} state relates to other properties of the material) is now clarified. My concern #1 consists of two parts: (1) the nature of the J_{eff} state (from a molecular cluster) – this is now properly discussed; and (2) how RIXS (a probe localized to a single atom) could explore the state of a molecule. This is not explicitly clarified earlier in the paper, but I would do with the new RIXS calculation (Fig. 5).

These being said, I still would not recommend the paper for publication, as I firmly believe the advances in this paper is incremental and do not live up to the standards of Nature Communications. The principle of using resonant X-rays to clarify the nature of J states is used in a series of papers in a range of journals (B. J. Kim et al., Science, 323, 1329 (2009); J. Kim et al., Phys. Rev. Lett. 108, 177003 (2012); S. Calder et al., Nature Communications 7: 11651 (2016); S. Calder et al., Phys. Rev. B 95, 020413(R) (2017)). As the authors agreed in the rebuttal letter, the J states is not a new notion, and the technical path was well laid out in previous papers. I could not agree with the “challenge and importance of the direct experimental observation” that the authors stated in the rebuttal letter as a reason for publication.

Reply:

We would like to thank Referee for the careful review and assessment. In particular, we are very glad to know that the main concerns are clearly addressed in the revised manuscript.

The magnetic resonant x-ray scattering (MRXS), which was used by Kim et al. in Science 323, 1329 (2009), can be used only for magnetically ordered materials. The $J_{\text{eff}}=3/2$ state cannot be clarified by MRXS whether the system is magnetically ordered or not. Here we demonstrated that RIXS technique can be useful even for a $J_{\text{eff}}=3/2$ system with no long-range magnetic order. Further, we found that all other works mentioned by Referee (i.e., Phys. Rev. Lett. by Kim et al. and two papers by Calder et al.) have nothing to do with the direct verification of J_{eff} state using the resonant X-rays technique. Thus we strongly believe that our current work carries a significant novelty from both technological and scientific point of view.